# Factors Influencing Men’s Attitudes toward HPV Vaccination in Males Included in the Chinese National Immunization Program

**DOI:** 10.3390/vaccines10071054

**Published:** 2022-06-30

**Authors:** Yi Tao, Huarui Shao, Ting Zhang, Junliang Pu, Chengyong Tang

**Affiliations:** 1Department of Phase I Clinical Trial Ward, The First Affiliated Hospital of Chongqing Medical University, Chongqing 400016, China; 204266@hospital.cqmu.edu.cn (Y.T.); pjl@stu.cqmu.edu.cn (J.P.); 2College of Pharmacy, Chongqing Medical University, Chongqing 400016, China; 2021120859@stu.cqmu.edu.cn; 3The First Clinical College, Chongqing Medical University, Chongqing 400016, China; 2021110290@stu.cqmu.edu.cn

**Keywords:** human papillomavirus, vaccination, HPV vaccine in males, factors influencing vaccination, national immunization programs

## Abstract

Background: Human papillomavirus (HPV) infection is the most common sexually transmitted disease, and it is associated with anogenital warts and oropharyngeal and anogenital cancers. Among female malignant tumors in China, the incidence of cervical cancer ranks second, with only breast cancer being more prevalent. HPV infection and related diseases affects both women and men. HPV vaccination is an optimal prevention strategy in preventing HPV infection and related diseases. The inclusion of the HPV vaccine in the national immunization program is an effective way to increase immunization coverage, reduce the burden of HPV related diseases, and increase national life expectancy. Objective: This study aimed to explore the factors influencing the attitudes of Chinese men toward the inclusion of the HPV vaccine in males included in the national immunization program, thus providing reference for launching the national immunization program policy. Methods: We invited men aged 20 to 45 to participate in an online survey. The participants were requested to complete a questionnaire, including sociodemographic characteristics, sexual behavior characteristics, knowledge of HPV and the HPV vaccine, and attitudes toward the HPV vaccine. A logistic regression model was constructed to analyze the influencing factors of attitudes. Results: A total of 660 males in China participated in this survey, and 80.45% supported the inclusion of HPV vaccines in national immunization programs. Participants earning CNY 100,000–200,000 (dds ratio (OR): 0.63, 95% confidence interval (CI): 0.39–1.00) or ≥200,000 (OR: 0.34, 95% CI: 0.17–0.68) were more likely to disapprove this strategy. Compared with people without a history of HPV infection, those with a history of HPV infection (OR: 1.84, 95% CI: 1.17–2.90) were more likely to approve. Men who had better knowledge of HPV were more likely to approve than men with less knowledge about HPV (OR: 1.44, 95% CI: 1.17–1.79). Compared with participants who did not know when the HPV vaccine should be given, those who knew that the ideal time of vaccination is before an individual becomes sexually active (OR: 1.75, 95% CI: 1.04–2.95) were more likely to approve. Conclusion: One in five men did not support the inclusion of HPV vaccines in national immunization programs, and they are likely to be from higher socioeconomic background and have poor knowledge of HPV. In order to implement comprehensive immunity, targeted actions need to be taken at national and public levels. In addition, when implementing measures, more attention needs to be paid to lower income men, men without a history of HPV infection and with poor knowledge of HPV, as well as young men.

## 1. Introduction

Human papillomavirus (HPV) belongs to the papillomaviridae family and is a small, double-stranded DNA virus classified as either low-risk HPVs (LR-HPVs, i.e., HPV6 and 11.) or high-risk HPVs (HR-HPVs, 16/18/31/33/35/39/45/51/52/56/58/59/66/68) [1,2]. LR-HPVs are associated with anogenital warts and skin warts, and HR-HPVs have been linked to oropharyngeal cancer (of the mouth, tonsils, and throat) and anogenital cancer [2]. Among them, the most carcinogenic is HPV 16/18, which causes more than 70% of cervical cancers and a high proportion of anogenital and oropharyngeal cancers [3]. HPV is the most common sexually transmitted infection, and it is estimated that about 80% of sexually active women and men will be infected with certain HPV types during their lifetime [4].

According to the World Health Organization (WHO), cervical cancer is the fourth most common cancer among women aged 15 to 44 in the world, with 604,000 and 342,000 worldwide cases of morbidity and death, respectively, in 2020 [5,6]. Statistics show that in 2020, there were about 110,000 new cases of cervical cancer and 59,000 deaths in China [5]. In addition to cervical cancer, HPV infection is also closely related to other cancers, including anal cancer (88%), vulvar cancer (43%), vaginal cancer (70%), penile cancer (50%), and oropharyngeal cancer (37%) [2,7]. Although the current research on HPV is focused on women, men are also susceptible to HPV infection and related diseases as malignant tumors (carcinoma of the anus, penis, and vulva) and benign lesions (genital warts and anus, mouth, and throat lesions). The global prevalence of HPV infection in men ranges from 5.3% to 42.2% [8], with China reporting a figure of about 10.5% [9].

Data show that HPV infection causes about 50.1% of penile cancers, of which HPV16 is the most common pathogenic type [10]. A total of 90% of anal cancers can be attributed to HPV infection, with a higher incidence among HIV-positive men and men who have sex with men (MSM) than in the general population [11]. HPV6 and HPV11 were associated with 90.0% of genital warts incidence [12] and 30% of oropharyngeal cancers are associated with HPV, with a four to five times higher incidence in men than in women [13].

The optimal prevention strategy for HPV infection is vaccination [14]. There are currently three licensed HPV vaccines (4-valent, 2-valent, and 9-valent) [15]. China has two domestically produced HPV vaccines and one bivalent vaccine (China) undergoing WHO prequalification. Global clinical studies have demonstrated that 4-valent and 9-valent HPV vaccines provide good protective effects in the male population [16,17,18,19] and both women and men vaccinated against HPV will contribute to the production of herd immunity effects [20]. According to statistics, HPV vaccination can prevent HPV-related cancers [12], including nearly 90% of cervical cancers, 22.8% of vulvar cancers, 24.5% of penile cancers, 66.7% of vaginal cancers, 79.0% of anal cancers, 21.3% of oropharyngeal cancers, 4.0% of oral cancers, and 2.7% of laryngeal cancers in the world [21]. HPV vaccines have been approved for male vaccination in many countries [22] but not in China. A previous study found that extending HPV vaccination to men is cost-effective and could further reduce the burden of HPV-related diseases for both sexes [23,24,25]. The Advisory Committee on Immunization Practices (ACIP) recommended that HPV vaccination be administered preferentially for men aged 13 to 21 who have not been vaccinated before, as well as for men who have sex with men and those who are immunocompromised [22]. By October 2020, 110 countries had included or partially included HPV vaccines in their national immunization programs [26], in which five countries with immunization coverage were as high as 90% [27]. At present, 11 vaccines are included in China’s national immunization program, including the hepatitis B vaccine, BCG vaccine, polio vaccine, DTP vaccine, etc. By vaccinating with the above 11 vaccines, 12 infectious diseases, such as hepatitis B, can be prevented.

At present, HPV vaccines are being evaluated in Chinese men as part of clinical trials. It is likely that the HPV vaccine will be licensed for use in males in China. The inclusion of the HPV vaccine in national immunization program is an effective way to increase immunization coverage, reducing the burden of disease, and increasing national life expectancy. In this study, we explored factors influencing the attitudes of Chinese men toward the HPV vaccine in males included in the national immunization program, thus providing reference for launching the national immunization program policy.

## 2. Methods

### 2.1. Study Design

Between 1 September and 30 November 2021, 678 men were recruited using a non-probability sampling method through WeChat in a phase I clinical trial ward of the first affiliated hospital of Chongqing medical university. A phase I clinical trial ward mainly conducts phase I clinical trials; thus, the participants were generally healthy people. There was a link to the survey website in the tweet, which was tweeted every 10 days from the ward’s WeChat account (most people who follow this account are healthy volunteers). In this way, men were invited to participate in an online survey to assess their attitudes toward the HPV vaccine for males included in the national immunization program. Detailed recruitment and study procedures were described in the tweet. All the participants were informed that the questionnaires would be answered anonymously, and participation in this study was voluntary. They were assured that their information would only be used for this study, and the answers of their questionnaires would have no influence on their career. To be eligible, participants needed to: (1) be aged 20 to 45; (2) be male (i.e., male sex at birth and male gender identity); (3) reside in China; and (4) have filled more than 95% of the questionnaire items.

Respondents who could not answer all the questions accurately were deemed ineligible. The survey took 4.2 min on average. To protect against fraudulent or duplicative enrollments, screening and survey responses were cross-referenced using age and IP addresses.

### 2.2. Measures

Socioeconomics—Characteristics included age (continuous), race/ethnicity (Han, Other), household registration (urban, rural), education level (junior college and below, undergraduate, postgraduate and above), marriage status (unmarried, married), occupation (public institutions, private enterprise, student, other), annual income (CNY; ≤100,000, 100,000–200,000, ≥200,000), and history of HPV infection (no, yes).

Sexual behavior—Participants were asked to report if they had a sexual partner in the last 6 months (no, yes), the number of sexual partners in the last 6 months (0, 1–2, 3–6, >6), frequency of condom use (never, occasionally, every time), and if they had ever attended a sexually transmitted disease examination (no, yes) or HIV testing (no, yes).

Knowledge of HPV—6 questions, including “Have you ever heard of human papillomavirus?” (no, yes) and “Can both men and women be infected with HPV?” (no, yes). The other 4 multiple choice questions were scored as the higher the score, the better the knowledge. Participants were asked to choose diseases associated with HPV infection, the high-risk factors of HPV, the transmission routes of HPV, and preventive methods of HPV infection. For the scoring of the assessment, participants achieved one point for one right answer and no points or one point deducted for a wrong answer. The total scores were divided into three levels by the quantile classification method, based on the values of the 25% quantile.

Knowledge of the HPV vaccine—Questions were asked as to whether the HPV vaccine can prevent malignancies (no, yes), when should the HPV vaccine be given (unknown, after first sexual activity, before first sexual activity), whether men can be vaccinated (no, yes), and if the HPV vaccine protects against all types of HPV infection (unknown, yes, no).

Attitudes toward the HPV vaccine—There were three variables regarding attitudes toward the HPV vaccine, as follows: “Would you like the HPV vaccine for males included in the national immunization program?” (disapproval, approval); “Do you prefer a domestic or imported HPV vaccine?” (indifferent, domestic, imported); and the price participants could afford to spend on an HPV vaccine (CNY; ≤1000, 1000–2000, 2000–3000, ≥3000).

### 2.3. Statistical Analyses

All statistical analyses were performed with SAS9.4 (Statistics Analysis System, version 9.4). For continuous variables conforming to a normal distribution, the values were expressed as the mean ± standard deviation (SD). Categorical variables were presented as frequencies and percentages. Groups were compared using two independent sample *t*-tests, differences in distributions for categorical variables were analyzed with the chi-square test. To determine the factors associated with men’s attitudes toward HPV vaccine for males being included in the national immunization program, an ordinal logistic regression model was constructed, in which stepwise regression was performed to choose variables included in the final model (*P*_entry_ = 0.05, *P*_stay_ = 0.05). All tests were performed by two-sided tests, with *p* < 0.05 as the statistical difference.

## 3. Results

In total, 678 participants answered the questionnaires, and 18 of them were excluded (17 participants were female, 1 participant did not finish the questionnaire, with missing items of over 95%). A final total of 660 participants were enrolled in this study.

### Socioeconomic Characteristics and Influencing Factors

A total of 660 men enrolled in this survey, of whom 531 approved of the male HPV vaccine being included in the national immunization program, and 129 men disapproved. The mean age of the participants was 28.04 ± 5.27. Participants were classified as being in the approval or disapproval group, based on their support of the vaccine for males being included in the national immunization program. The results were displayed in Table 1, household registration, marriage status, annual income, and history of HPV infection, showed statistical differences between the approval and disapproval groups. In the approval group, there were 314 (59.13%) urban participants, 382 (71.94%) were unmarried, 305 (57.44%) participants had an annual income of CNY ≤ 100,000, and 236 (44.44%) had a history of HPV infection. In the disapproval group, there were 90 (69.77%) urban participants, 86 (66.67%) were unmarried, 58 (44.96%) participants had an annual income of CNY ≤ 100,000, and 34 (26.36%) had a history of HPV infection.

Regarding sexual behavior, the approval and disapproval groups showed statistical differences. In total, 481 (72.88%) participants had had a sexual partner in the last 6 months, 379 of whom were in the approval group and 102 in the disapproval group (Table 2).

Among the variables regarding knowledge of HPV (Table 3), the scores related to diseases associated with HPV, high-risk factors of HPV infection, transmission routes of HPV, and preventive methods of HPV infection showed statistical differences between the approval and disapproval groups.

The variables regarding knowledge of the HPV vaccine were shown in Table 4. Participants’ knowledge of when the HPV vaccine should be given was statistically different between the approval and disapproval groups. There were 501 (75.91%) participants who answered that the best time for the vaccination to be administered is before the first sexual activity, 414 (77.97%) of whom approved for the vaccine being included in national immunization program and 87 (67.44%) disapproved.

No statistical differences were observed in attitudes toward the HPV vaccine between the groups. In the approval and disapproval groups, 276 (51.98%) and 71 (55.04%) participants had no preference between domestic or imported HPV vaccines, respectively, 153 (28.81%) and 41 (31.78%) participants, respectively, preferred an imported one. The prices participants stated they could afford the HPV vaccine were close between the approval and disapproval groups. Comparisons of participants’ preferences for vaccine prices and types are shown in Figure 1.

A multivariate logistic regression analysis was conducted to identify potential associated factors that may be associated with men’s attitudes toward the HPV vaccine being administered to males in the national immunization program, and the results were shown in Table 5; the variable selection process yielded 10 variables closely related to these attitudes. Of the 10 variables, household registration, annual income, education level, marriage, history of HPV infection, existence of a sexual partner in the last 6 months, diseases associated with HPV, high-risk factors of HPV, preventive methods of HPV infection, and timing of the HPV vaccine were the related factors.

The 10 variables that provided significance in the univariate analyses were chosen as independent variables to be included in the multivariate logistic regression analysis. As a result, a total of four factors entered the logistic regression equation; the maximum contribution was annual income, followed by timing of the HPV vaccine, high-risk factors of HPV, and history of HPV infection (Table 5). Comparing participants with annual income of CNY ≤ 100,000, those earning CNY 100,000–200,000 (odds ratio (OR): 0.63, 95% confidence interval (CI): 0.39–1.00) or CNY ≥ 200,000 (OR: 0.34, 95% CI: 0.17–0.68) were more likely to disapprove. Compared with participants without a history of HPV infection, those with a history of HPV infection (OR: 1.84, 95% CI: 1.17–2.90) were more likely to approve. For the sores related to the high-risk factors of HPV (OR: 1.44, 95% CI: 1.17–1.79), the higher the scores were, the more likely approval was. Compared with those who did not know when the HPV vaccine should be given, participants were more likely to approve if they believed the vaccine should be administered after first sexual activity (OR: 5.51, 95% CI: 0.67–45.10) or before the first sexual activity (OR: 1.75, 95% CI: 1.04–2.95).

## 4. Discussion

This study explored the factors influencing Chinese men’s attitudes toward the HPV vaccine being included in the national immunization program. HPV infection is related to a variety of human cancers; the HPV vaccine is a cancer-preventive vaccine that has been shown to be effective against HPV induced cancers. Meanwhile, although male HPV vaccination is becoming a priority in some countries, it is currently unavailable to males in China.

The study results showed that 80.45% of men in our survey supported the inclusion of HPV vaccines in national immunization programs, and the influencing factors were annual income, history of HPV disease, knowledge of the high-risk factors for HPV infection, and the optimal timing of HPV vaccination. Participants of lower socioeconomic background were more supportive of the inclusion of HPV vaccines in national immunization programs. Previous studies found that the cost of HPV vaccination was the main factor hindering vaccination [28,29]. The cost of three doses of 9-valent HPV vaccine in China was around CNY 3000, which was a particular economic burden for many people, especially those with low incomes. The high-income group could afford the vaccines and had less desire for the vaccine to be included in national immunization programs. Female vaccination results showed that the low female HPV vaccination coverage rate in China was connected with the high cost of vaccines. Therefore, including HPV vaccines in national immunization programs is an effective way to improve vaccination coverage. Men with a history of HPV infection were more supportive of the vaccine. Perhaps as a result of better understanding the dangers of HPV infection, these participants were more willing to be vaccinated to prevent or reduce the risk of HPV infection and related diseases. Natural HPV immunity produced by the associated antibody and preventing re-infection of the same type of HPV is difficult to achieve [30,31]. However, the HPV vaccine has a significant protective effect in people with recurrent (transient or persistent) HPV infection of previous vaccine types. Therefore, HPV vaccination is recommended for people of appropriate age, regardless of the presence of HPV infection or cytological abnormalities [32,33,34]. Participants who knew more about the risk factors of HPV infection revealed a more supportive attitude and were more eager to be vaccinated to prevent infection, due to their increased awareness of people’s susceptibility to HPV and the adverse consequences of infection [35]. People who knew when the HPV vaccine should be administered were more aware of the benefits of vaccines: the earlier the better, within the approved age range [36]. Therefore, more attention should be paid to young men when taking targeted actions.

Previous studies found that people’s willingness to use HPV vaccines was related to their level of education [37]. In this study, the level of education had no influence on men’s attitude toward the HPV vaccine being administered to males included in the national immunization program. This was possibly due to the current public awareness of vaccinations being generally high, and the internet making knowledge of vaccines more comprehensive. In addition, high-risk factors of infection, such as the number of sexual partners and the existence of a sexual partner in the last 6 months, did not affect attitudes, and they were not independent influencing factors of attitudes. Nevertheless, having multiple sexual partners and frequent sex are high-risk factors for HPV infection, meaning that people with such behaviors would benefit more from HPV vaccination [38,39].

The evidence of the cost-effectiveness of extending HPV vaccination to boys is ambiguous and has not verified worldwide [40], while researches claimed that a gender-neutral strategy would be cost-effective (or marginally cost-effective) in some countries [23,24,25,41]. Before adoption the strategy of including the male HPV vaccine in national immunization program, China should carry out further cost-effectiveness analyses. In addition to cost-effectiveness, safety is an important consideration when promoting HPV vaccine use and inclusion in national immunization programs. Studies have shown that the HPV vaccine is safe and well tolerated in both men and women [42]. To summarize, HPV vaccination is potentially a cost-effective preventive measure, and actions should be taken to integrate vaccines into immunization programs and increase vaccination rates in the population. In order to carry out this policy, recommendations are proposed as follows. Firstly, reports in the literature have stated that doctors’ advice is an effective way to promote vaccination [43]. Health science education promoting HPV knowledge should be strengthened through schools and hospitals to increase public confidence in the vaccines, and, at the same time, HPV-related science videos and information could be promoted through the media. This education strategy should be applied not only for women but also for men. HPV prevention efforts should focus on both the promotion of condom use and HPV vaccination. Consistent condom use was verified to reduce the risk of HPV infection [44]. Secondly, effective measures include accelerating the review and approval process of HPV vaccines in China, improving the enthusiasm of manufacturers and suppliers to ensure vaccine supply and, more importantly, reducing the price of vaccines to improve the accessibility and affordability of HPV vaccines. Thirdly, HPV and cervical cancer testing for women is widely available but is not available for men in most institutions. Providing HPV screening for men, especially for those high-risk groups, such as men who have sex with men (MSM), could be further prioritized. The existing screening methods mainly include Pap smears, liquid-based cytology tests, and HPV tests. Cytological specimens through swabs of the penis, scrotum, and anus are mainly used [45]. The collection of cytological specimens depends on doctors and is invasive. First-void urine HPV screening has the advantages of being simple, convenient, and non-invasive, making it the most acceptable of all diagnostic methods; it does not affect the body’s HPV infection status and can be readily offered as a screening and diagnostic pre-vaccination test for both sexes [46,47]. Lastly, a lack of vaccine information is also one of the reasons hindering vaccination [48]. Modern information technology should be developed to release vaccine-related information to improve vaccine accessibility and establish a connection between HPV vaccination and related HPV screening results in order to achieve an early prevention, early diagnosis, and early treatment strategy.

## 5. Conclusions

In summary, this study illustrated the factors influencing men’s attitudes toward the HPV vaccine administered to males being included in the national immunization program. The findings indicate that although most men approve of the HPV vaccine being included in the immunization program, one in five men do not support this measure. In order to implement a comprehensive immunization policy, targeted actions should be taken at national and public levels. In addition, more attention needs to be paid when implementing strategies to those who are younger, with lower income, without a history of HPV infection, and with poor knowledge of HPV. Lastly, HPV vaccination and screening combined with relevant promotion and education strategies should be implemented to prevent HPV infection and eliminate HPV-induced cancers in the coming decades.

## 6. Limitations

The notable limitations of this study were as follows. Firstly, although a variety of factors were found to be related to men’s attitudes toward HPV vaccination being included in the national immunization program, the design of the questionnaire was not comprehensive enough and potential variables may have been ignored, i.e., sexual orientation may be a significant factor, but it was not included in the questionnaire. Secondly, the recall bias and social expectation bias cannot be eliminated in a self-reported questionnaire. Thirdly, we sent the questionnaire link via the WeChat account of a phase I clinical trial ward. The participants viewing this account may be more interested in health topics, more eager to participate in an HPV vaccine clinical trial, and their knowledge of HPV and vaccines may be higher than the population average; this may limit the generalizability of the study results.

## Figures and Tables

**Figure 1 vaccines-10-01054-f001:**
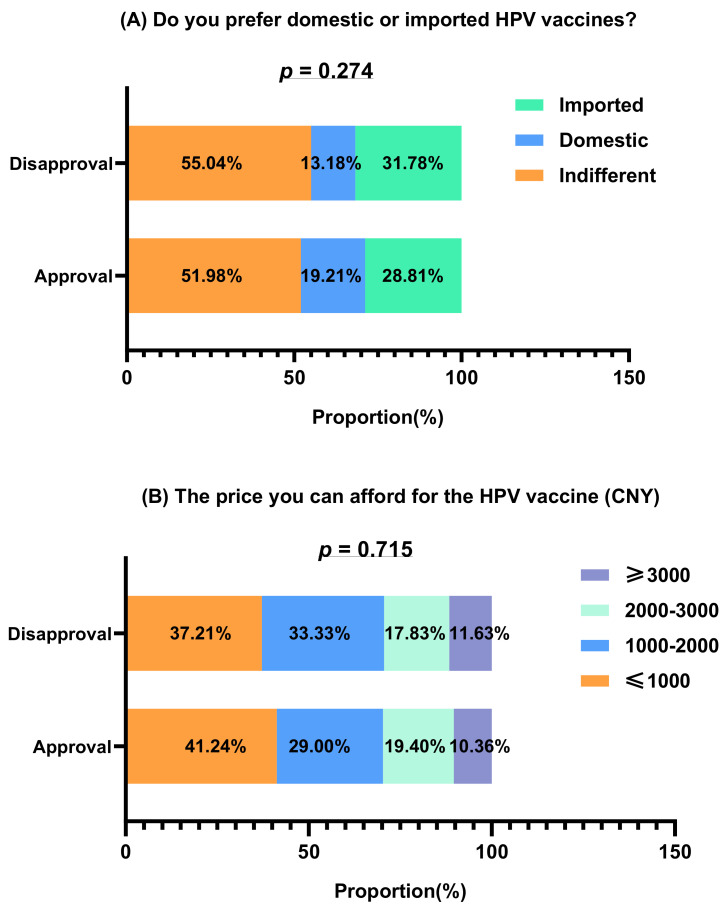
(**A**) Comparison of preferences for domestic or imported vaccines. (**B**) Comparison of affordable vaccine prices for participants (*n* = 660).

**Table 1 vaccines-10-01054-t001:** Socioeconomic characteristics.

Variable	Total (*n* = 660)	Approval (*n* = 531)	Disapproval (*n* = 129)	*p*
Age	28.04 ± 5.27	27.94 ± 5.32	28.43 ± 5.05	0.878
Ethnicity				0.451
Han	623 (94.39%)	503 (94.73%)	120 (93.02%)	
Other	37 (5.61%)	28 (5.27%)	9 (6.98%)	
Household registration				0.032
Urban	404 (61.21%)	314 (59.13%)	90 (69.77%)	
Rural	256 (38.79%)	217 (40.87%)	39 (30.23%)	
Education level				0.651
Junior college and below	171 (25.91%)	140 (26.37%)	31 (24.03%)	
Undergraduate	320 (48.48%)	259 (48.78%)	61 (47.29%)	
Postgraduate and above	169 (25.61%)	132 (20.72%)	37 (28.68%)	
Occupation				0.434
Public institutions	307 (46.52%)	258 (48.59%)	49 (37.98%)	
Private enterprise	81 (12.27%)	70(13.18%)	11 (8.53%)	
Student	150 (22.73%)	134 (25.24%)	16 (12.40%)	
Other	83 (12.58%)	69 (12.99%)	14 (10.85%)	
Marriage				0.027
Unmarried	468 (70.91%)	382 (71.94%)	86 (66.67%)	
Married	192 (29.09%)	149 (28.06%)	43 (33.33%)	
Annual income				0.014
CNY ≤ 100,000	363 (55.00%)	305 (57.44%)	58 (44.96%)	
CNY 100,000–200,000	234 (35.45%)	182 (34.27%)	52 (40.31%)	
CNY ≥ 200,000	63 (9.55%)	44 (8.29%)	19 (14.73%)	
History of HPV infection				0.000
No	390 (59.09%)	295 (55.56%)	95 (73.64%)	
Yes	270 (40.91%)	236 (44.44%)	34 (26.36%)	

Data are expressed as *n* (%), mean ± SD, *p* < 0.05. Statistical analysis was performed using the *t*-test and chi-square test.

**Table 2 vaccines-10-01054-t002:** Sexual behavior.

Variable	Total (*n* = 660)	Approval (*n* = 531)	Disapproval (*n* = 129)	*p*
Number of sexual partners				0.132
0	179 (27.12%)	152 (28.63%)	27 (20.93%)	
1–2	466 (70.61%)	367 (69.11%)	99 (76.74%)	
3–6	9 (1.36%)	7 (1.32%)	2 (1.55%)	
>6	6 (0.91%)	5 (0.94%)	1 (0.78%)	
Sexual partner in the last 6 months				0.047
No	179 (27.12%)	152 (28.63%)	27 (20.93%)	
Yes	481 (72.88%)	379 (71.37%)	102 (79.07%)	
Frequency of condom use				0.160
Frequently	367 (55.61%)	295 (55.56%)	72 (55.81%)	
Occasionally	83 (12.58%)	61 (11.49%)	22 (17.05%)	
Never	210 (31.82%)	175 (32.96%)	35 (27.13%)	
STD examination				0.791
No	487 (73.79%)	393(74.01%)	94 (72.87%)	
Yes	173 (26.21%)	138 (25.99%)	35 (27.13%)	
HIV testing				0.876
No	431 (65.30%)	346 (65.16%)	85 (65.89%)	
Yes	229 (34.70%)	185 (34.84%)	44 (34.11%)	

Data are expressed as *n*(%), *p* < 0.05. Statistical analysis was performed using the chi-square test.

**Table 3 vaccines-10-01054-t003:** Knowledge of HPV.

Variable	Total (*n* = 660)	Approval (*n* = 531)	Disapproval (*n* = 129)	*p*
Have you ever heard of HPV?				0.131
No	55 (8.33%)	40 (7.53%)	15 (11.63%)	
Yes	605 (91.67%)	491 (92.47%)	114 (88.37%)	
Both men and women can be infected with HPV				0.819
No	78 (11.82%)	62 (11.68%)	16 (12.40%)	
Yes	582 (88.18%)	469 (88.32%)	113 (87.60%)	
Diseases associated with HPV infection (scores)				<0.001
0–1	84 (12.73%)	54 (10.17%)	30 (23.26%)	
2–3	335 (50.76%)	275 (51.79%)	60 (46.51%)	
4	241 (36.52%)	202 (38.04%)	39 (30.23%)	
High-risk factors of HPV (scores)				<0.001
1–2	146 (22.12%)	95 (17.89%)	51 (39.53%)	
3	192 (29.09%)	155 (29.19%)	37 (28.68%)	
4–6	322 (48.79%)	281 (52.92%)	41 (31.78%)	
Transmission routes of HPV (scores)				0.003
1	153 (23.18%)	114 (21.47%)	39 (30.23%)	
2	343 (51.97%)	271 (51.04%)	72 (55.81%)	
3	164 (24.85%)	146 (27.50%)	18 (13.95%)	
Preventive methods of HPV infection F (scores)				0.003
1–2	76 (11.52%)	53 (9.98%)	23 (17.83%)	
3–4	242 (36.67%)	187 (35.22%)	55 (42.64%)	
5–6	342 (51.82%)	291 (54.80%)	51 (39.53%)	

Data are expressed as *n* (%), *p* < 0.05. Statistical analysis was performed using the chi-square test.

**Table 4 vaccines-10-01054-t004:** Knowledge of HPV vaccine.

Variable	Approval (*n* = 531)	Disapproval (*n* = 129)	Total (*n* = 660)	*p*
Can the HPV vaccine prevent malignancies?				0.761
No	41 (7.72%)	11 (8.53%)	41 (7.88%)	
Yes	490 (92.28%)	118 (91.47%)	490 (92.12%)	
When should the HPV vaccine be given?				0.005
Unknown	103(19.40%)	41 (31.78%)	144(21.82%)	
After first sexual activity	14 (2.64%)	1 (0.78%)	15 (2.27%)	
Before first sexual activity	414 (77.97%)	87 (67.44%)	501 (75.91%)	
Can men be vaccinated?				0.521
No	28 (5.27%)	9 (6.98%)	37 (5.61%)	
Yes	503 (94.73%)	120 (93.02%)	623 (94.39%)	
Does the HPV vaccine protect against all types of HPV infection?				0.287
Unknown	83 (15.63%)	26 (20.16%)	109 (16.52%)	
Yes	73 (13.75%)	21 (16.28%)	94 (14.24%)	
No	375 (70.62%)	82 (63.57%)	457 (69.24%)	

Data are expressed as *n* (%), *p* < 0.05. Statistical analysis was performed using the chi-square test.

**Table 5 vaccines-10-01054-t005:** Multivariable analysis of factors influencing men’s attitudes toward HPV vaccine in males being included in the national immunization program.

Factors	β-Coefficient	OR (95%CI)	*p*
Household registration			0.142
Urban	reference	1	
Rural	0.34	1.40 (0.89–2.20)	
Education level			0.189
Junior college and below	reference	1	
Undergraduate	−0.02	0.98 (0.57–1.70)	0.950
Postgraduate and above	−0.46	0.63 (0.34–1.18)	0.149
Marriage status			0.331
Unmarried	reference	1	
Married	0.24	1.27 (0.78–2.07)	
Annual income (CNY)			0.007
≤100,000	reference	1	
100–200,000	−0.47	0.63 (0.39–1.00)	0.051
≥200,000	−1.08	0.34 (0.17–0.68)	0.002
History of HPV infection			0.008
No	reference	1	
Yes	0.61	1.84 (1.17–2.90)	
Sexual partner in the last 6 months			0.233
No	reference	1	
Yes	−0.32	0.73 (0.43-1.23)	
Diseases associated with HPV infection	0.12	1.13 (0.91–1.39)	0.269
High-risk factors of HPV	0.37	1.44 (1.17–1.79)	0.001
Transmission routes of HPV	0.10	1.10 (0.78–1.56)	0.574
Preventive methods of HPV infection	−0.09	0.91 (0.75–1.11)	0.363
Timing of vaccine administration			0.049
Unknown	reference	1	
After first sex	1.71	5.51 (0.67–45.10)	0.112
Before first sex	0.56	1.75 (1.04–2.95)	0.036

CI, confidence interval; OR, odds ratio.

## Data Availability

The data presented in this study are available on request from Huarui Shao. The data are not publicly available due to privacy of participants.

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
