# Peer review of "Factors Influencing Men’s Attitudes toward HPV Vaccination in Males Included in the Chinese National Immunization Program"

_vaccines, 2022, doi:10.3390/vaccines10071054_

Round 1
Reviewer 1 Report
Dear authors your article «Factors influencing men's attitudes toward HPV vaccine in males included in national immunization program» is well presented and written and I have recommended that the journal's editor consider to accept your article – pending minor revisions.
However, you need to comply with each element of the report so that you may get your article published at Vaccines.
The manuscript is well written and I believe it can capture the reader's attention early. the authors may expand on 3 main domains
1. Please add a paragraph describing the positive effect of HPV vaccination on individuals (women and possible men) with established HPV infection and plausible regression of infection. This issue could influence individuals (having already acquired HPV infection or not) to accept vaccination.
2. Please add some sentences on the HPV screening methods possible available for men. Urine HPV screening with first void seems to be the most acceptable of all diagnostic approaches and can be easily offered in both sexes as screening and diagnostic pre-vaccination test.
3. Please discuss, based on sexual lifestyle characteristics that may alter HPV positivity, the protective effect in transmission and HPV clearance of condoms as well as the limitations in condom use consistency.
Author Response
Dear reviewer,
Thank you for your comments concerning our manuscript entitled “Factors Influencing Men’s Attitudes Toward HPV Vaccine in Males Included in National Immunization Program” (ID: vaccines-1735075). We thank the editors and reviewers for your careful read and thoughtful comments on our manuscript. We have addressed comments and made correction. The corrections in the paper and the respondents to the reviewer’s comments are as followings.
Responds to the reviewer’s comments:
Answers to reviewer1:
- Please add a paragraph describing the positive effect of HPV vaccination on individuals (women and possible men) with established HPV infection and plausible regression of infection. This issue could influence individuals (having already acquired HPV infection or not) to accept vaccination.
We have added contents about the positive effect of HPV vaccination on individuals with established HPV infection and plausible regression of infection from lines 300 to lines 305.
- Please add some sentences on the HPV screening methods possible available for men. Urine HPV screening with first void seems to be the most acceptable of all diagnostic approaches and can be easily offered in both sexes as screening and diagnostic pre-vaccination test.
We have added sentences about the HPV screening methods possible available for men as urine HPV screening, from lines 350 to lines 357.
- Please discuss, based on sexual lifestyle characteristics that may alter HPV positivity, the protective effect in transmission and HPV clearance of condoms as well as the limitations in condom use consistency.
We have added the protective effect of condoms use against HPV transmission, from lines 342 to lines 344.
We appreciate for editors and reviewers’ warm work earnestly, and hope that the corrections will meet with approval. Once again, thank you very much for your comments and suggestions.

Reviewer 2 Report
The research manuscript entitled “Factors influencing men’s attitudes towards HPV vaccine in males included in national immunization program” by Tao et al. is an outstanding piece of research and this is really an unexplored research topic so far. Without a doubt, it is a highly suitable manuscript for publication in Vaccines. However, the author needs to address the following comments before being accepted for publication
Highly recommended for revision of title of the manuscript. the suggested title “HPV vaccination: Influence of Men’s Attitude”
In keywords author can add, vaccination, factor influencing vaccination, and the full name of HPV
The author should disclose the abbreviation at their first appearance in the manuscript. eg. HPV in the introduction did not disclose. There are several similar issues throughout the manuscript
The author should cite the following literature in the introduction
https://www.sciencedirect.com/science/article/abs/pii/S1773224722002611
line 49: it should be “World Health Organization (WHO)”
in all tables: the last column “p” needs to be disclosed the details below the table
in line 164: it should be 484 (73.33%). There is a space between the number and the start of the bracket. Similar errors available in the manuscript and in the tables throughout author need to be very careful and rectify the mistakes
in table 6: what is the meaning of OR, need to disclose below the table
conclusion significantly needs improvement
authors need to add at least one scheme and 3-5 figures. The author can also try to modify the tables into figures if possible.
Author Response
Dear reviewer,
Thank you for your comments concerning our manuscript entitled “Factors Influencing Men’s Attitudes Toward HPV Vaccine in Males Included in National Immunization Program” (ID: vaccines-1735075). We thank the editors and reviewers for your careful read and thoughtful comments on our manuscript. We have addressed comments and made correction. The corrections in the paper and the respondents to the reviewer’s comments are as followings.
Responds to the reviewer’s comments:
Answers to reviewer2:
- Highly recommended for revision of title of the manuscript. the suggested title “HPV vaccination: Influence of Men’s Attitude”
Thanks for your suggestion, this study aimed to explore the factors influencing attitudes of Chinese men toward the inclusion of HPV vaccine in males included in national immunization program.
- In keywords author can add, vaccination, factor influencing vaccination, and the full name of HPV
Keywords was added from lines37 to lines38 .
- The author should disclose the abbreviation at their first appearance in the manuscript. eg. HPV in the introduction did not disclose. There are several similar issues throughout the manuscript
We checked the full text, abbreviations at their first appearance were added.
- The author should cite the following literature in the introduction https://www.sciencedirect.com/science/article/abs/pii/S1773224722002611
Thanks for your suggestion, the above literature was cited in the introduction.
- line 49: it should be “World Health Organization (WHO)”
Thanks for your comment; it has been revised in line 53.
- in all tables: the last column “p” needs to be disclosed the details below the table
“P<0.05” and notes were disclosed below the tables.
- in line 164: it should be 484 (73.33%). There is a space between the number and the start of the bracket. Similar errors available in the manuscript and in the tables throughout author need to be very careful and rectify the mistakes
Thanks for your comment. We checked the full text, a space between the number and the start of the bracket was added.
- In table 6: what is the meaning of OR, need to disclose below the table
“OR, odds ratio” was disclosed below the table6.
- Conclusion significantly needs improvement
Conclusion was improved, we look forward to your further valuable advices.
- Authors need to add at least one scheme and 3-5 figures. The author can also try to modify the tables into figures if possible.
Thanks for your comment. Two figures were added (figure1 and figure2).
We appreciate for editors and reviewers’ warm work earnestly, and hope that the corrections will meet with approval. Once again, thank you very much for your comments and suggestions.

Reviewer 3 Report
Dear authors,
the work is well done and highlight the problem linked to the vaccination acceptance in a modern country and how to resolve the problem. This can be used also for improve the covid 19 vaccination.
Author Response
Dear reviewer,
Thank you for your comments concerning our manuscript entitled “Factors Influencing Men’s Attitudes Toward HPV Vaccine in Males Included in National Immunization Program” (ID: vaccines-1735075). We thank the editors and reviewers for your careful read and thoughtful comments on our manuscript. We have addressed comments and made correction. The corrections in the paper and the respondents to the reviewer’s comments are as followings.
Responds to the reviewer’s comments:
Answers to reviewer3:
Thanks for your comments, the revised paper is attached, thank you very much for your further comments and suggestions.
We appreciate for editors and reviewers’ warm work earnestly, and hope that the corrections will meet with approval. Once again, thank you very much for your comments and suggestions.

Reviewer 4 Report
Tao et. al., manuscript on HPV vaccine in males addresses an important issue in cancer treatment, and their study aimed to investigate the factors influencing attitudes of Chinese men toward the inclusion of HPV vaccine in males included in national immunization program, thus providing references for launching the national immunization program policy. However, the content is a very hot and promising topic, but the manuscript is not written at that level, and it has many drawbacks and insufficient data. The study design and questioner are inadequate. In the current study, the platform used for the questioner is untrustworthy. In their questioner, the author should also discuss the vaccine's side effects, etc.. According to the CDC, more than 67 million doses of the HPV vaccine were administered nationally between June 2006 and March 2014. There were 25,176 reports of adverse events to the Vaccine Adverse Event Reporting System (VAERS). Author need to discuss about all this points. The author only mentioned one point in the results section, 3.1 Socioeconomic Characteristics and Influencing Factors; more points should be mentioned in the result section; no need to add extra subdivisions. What does the author believe is the problem? Rather than vaccination, social awareness is required, as it is well known that HVP is transmitted sexually, saving the poor population money on vaccination. Many points are missing in this manuscript, which I believe is insufficient and poorly organized, and it should be rewritten with additional data, additional questioners, and a lot of relevant discussion. I sincerely apologize for rejecting the paper.
Author Response
Dear reviewer,
Thank you for your comments concerning our manuscript entitled “Factors Influencing Men’s Attitudes Toward HPV Vaccine in Males Included in National Immunization Program” (ID: vaccines-1735075). We thank the editors and reviewers for your careful read and thoughtful comments on our manuscript. We have addressed comments and made correction. The corrections in the paper and the respondents to the reviewer’s comments are as followings.
Responds to the reviewer’s comments:
Answers to reviewer4:
- The content is a very hot and promising topic, but the manuscript is not written at that level, and it has many drawbacks and insufficient data.
Thanks for your comments, we have revised the paper and checked the data, we are looking forward to your further comments and suggestions.
- The study design and questioner are inadequate. In the current study, the platform used for the questioner is untrustworthy. In their questioner, the author should also discuss the vaccine's side effects, etc.
Thanks for your valuable suggestions, the drawbacks of questionnaire design were limitations of this study, we will make it more comprehensively in further study. The vaccine's side effects were added in discussion from lines 327 to lines 333.
- Author need to discuss about all this points. The author only mentioned one point in the results section, 3.1 Socioeconomic Characteristics and Influencing Factors; more points should be mentioned in the result section; no need to add extra subdivisions.
Thanks for your valuable suggestions. We have enriched the content of the discussion section. In results section, we thought that extra subdivisions can make the consequences more clear. We described more about part 3.1, because socioeconomic characteristics were the overall situation of the participants. For the results without statistical differences in the tables we described less to avoid repetitive verbosity.
- What does the author believe is the problem? Rather than vaccination, social awareness is required, as it is well known that HVP is transmitted sexually, saving the poor population money on vaccination. Many points are missing in this manuscript, which I believe is insufficient and poorly organized, and it should be rewritten with additional data, additional questioners, and a lot of relevant discussion.
Thanks for your valuable suggestions. From a disease prevention perspective, we thought that HPV vaccination, screening combined with relevant propaganda and education were the key points to prevent HPV infection and eliminate HPV induced cancers in the coming decades. However, the optimal prevention strategy for HPV infection is vaccination. The purpose of this study was analyzing the factors influencing attitudes of Chinese men toward the HPV vaccine in males included in national immunization program. According to the study results, more attentions need to be paid to those men with lower income, without HPV infection history, with low Knowledge of HPV and in younger age when taking targeted actions at national and public levels.
We appreciate for editors and reviewers’ warm work earnestly, and hope that the corrections will meet with approval. Once again, thank you very much for your comments and suggestions.

Round 2
Reviewer 2 Report
The thoroughly revised manuscript recommended for publication
Author Response
Responds to reviwer:
Thanks for your approval, the full text was carefully revised, and English of article was edited. Once again, thank you very much for your comments and suggestions.
Reviewer 4 Report
Sorry to say, but the arguments and modifications are inadequate, and it should be rewritten with a different experimental setup and analysis. I regret having to reject this paper again because the methodology and conclusion are both questionable.
Author Response
Dear editor,
Thank you for your comments concerning our manuscript entitled “Factors Influencing Men’s Attitudes Toward HPV Vaccine in Males Included in National Immunization Program” (ID: vaccines-1735075). We thank you for your careful read and thoughtful comments on our manuscript. We have addressed comments and made correction. The corrections in the paper and the respondents to the reviewer’s comments are as followings.
Responds to comments:
Thanks for your comment, the sections of discussion and conclusion were revised, and English of article was edited.
We hope that the corrections will meet with approval. Once again, thank you very much for your comments and suggestions.